



# Anticipating a risky future: LSTM models for spatiotemporal extrapolation of population data in areas prone to earthquakes and tsunamis in Lima, Peru

Christian Geiß[1,2], Jana Maier[1], Emily So[3], Elisabeth Schoepfer[1], Sven Harig[4], Juan Camilo Gómez Zapata[5,6] and Yue Zhu[3,7]

[1]German Aerospace Center (DLR), German Remote Sensing Data Center (DFD), Wessling, 82234, Germany
[2]Department of Geography, University of Bonn, Bonn, 53115, Germany
[3]Centre for Risk in the Built Environment, University of Cambridge, CB2 1PX, Cambridge, UK
[4]Alfred Wegener Institute, Helmholtz Centre for Polar and Marine Research (AWI), Bremerhaven, 27570, Germany
[5]Helmholtz Centre Potsdam GFZ German Research Centre for Geosciences, Potsdam, 14473, Germany
[6]Institute for Geosciences, University of Potsdam, Potsdam, 14476, Germany
[7]Institute of Environmental Engineering, ETH Zurich, Zurich, 8093, Switzerland

*Correspondence to*: Christian Geiß (christian.geiss@dlr.de)

**Abstract.** In this paper, we anticipate geospatial population distributions to quantify the future number of people living in earthquake-prone and tsunami-prone areas of Lima and Callao, Peru. We capitalize upon existing gridded population time series data sets, which are provided on an open source basis globally, and implement machine learning models tailored for time series analysis, i.e., Long Short-Term Memory-based (LSTM) networks, for prediction of future time steps. In detail, we harvest WorldPop population data and learn LSTM and Convolutional LSTM models equipped with both unidirectional and bidirectional learning mechanisms and derived from different feature sets, i.e., driving factors. To gain insights regarding the competitive performance of LSTM-based models in this application context, we also implement multilinear regression and Random Forest models for comparison. The results clearly underline the value of the LSTM-based models for forecasting gridded population data. The best trained model is deployed for anticipation of population along a three-year interval until the year 2035. Especially in areas of high peak ground acceleration of $207 - 210 \frac{m}{s^2}$, the population will experience a growth of almost 30 % over the forecasted time span which simultaneously corresponds to 70 % of the predicted additional inhabitants of Lima. The population in the tsunami inundation area will grow by 61 % until 2035, which is substantially more than the average growth of 35 % for the city. Uncovering those relations can help urban planners and policy makers to develop effective risk mitigation strategies.

## 1 Introduction

Natural disasters represent a perpetual peril to mankind. Such events frequently result in substantial losses. The anticipated growth of the world population with a peak of 9.7 billion people in the year 2050 (United Nations, 2022) is expected to expose more people to natural hazards than ever before (Iglesias et al., 2021; Cremen et al., 2022). The dynamic change of geospatial population distributions due to both population growth and urbanization processes (UN Habitat, 2016), thus, induces the need to constantly update and accurately anticipate future geospatial population distributions in hazard-prone areas. Such approach enables urban planners and policymakers to develop effective strategies for risk mitigation. This need is also embedded in the UN Sendai Framework for Disaster Risk Reduction, which explicitly stresses the importance of preparing for future disasters via strategies that minimize uncontrolled settlement development in areas at peril (UNISDR, 2015).

As a key variable to characterize natural hazard-related exposure, obtaining geospatial data on population distribution is essential. To anticipate future geospatial population distributions, different families of methods can be considered generally: Rule-based methods establish a set of explicitly defined rules for transition trajectories over time. This family of methods contains i) Cellular Automata techniques (Clarke, 2014) which represent discrete spatiotemporal dynamic systems based on



local rules; ii) Agent-based Modelling which simulates dynamic interactions among agents in a virtual environment (Abar et al., 2017); iii) Markov Chain Models which represent a stochastic process that produces sequential states in which each prediction is dependent on the previous state (Gagniuc, 2017).

However, especially recently, a second family of methods, i.e., techniques of Machine Learning (ML), were also utilized for
predicting transition trajectories in the context of population modeling. The underlying idea is to infer a decision rule (e.g., a function) from properly encoded prior knowledge (i.e., labeled training samples) related to time series data to predict changes (Zhu, 2023). For instance, Chen et al. (2020) integrate historical population maps and multiple machine learning algorithms, i.e., XGBoost, Random Forest (RF), and a multi-layer perceptron neural network, to predict future built-up land and population distributions. Kubota et al. (2022) implemented a Graph Convolutional Network for short-term population prediction based
on population count data collected through mobile phone signals. Zheng and Zhang (2020) implement a Convolutional LSTM (ConvLSTM) network for weekly population distribution prediction based on geolocated social media data, i.e., Tencent positioning data.

Generally, earth observation data were recognized to be very valuable to measure changes on the land surface in a spatially continuous way over long time frames (Koehler and Künzer, 2020). Such data sets were used in combination with advanced
ML techniques to anticipate land-use and land-cover expansion (Zhu et al., 2021 a, b; Wang et al., 2022). By integrating earth observation data, different initiatives offer continuous gridded geospatial population data over a long time frame: WorldPop (Lloyd et al., 2017; Stevens et al., 2015), and LandScan (Dobson et al., 2000) provide yearly geospatial population estimates starting in the year 2000. The data sets are created with a top-down approach by disaggregating census information based on earth observation imagery and ancillary spatial covariates. In this study, *from a data-oriented perspective*, we mitigate the
often expensive process of compiling time series data through innovatively make use of existing time series global population data sets, which are provided on an open source basis, to anticipate future geospatial population distributions along a three-year interval up to the year 2035.

*From a methodological point of view*, we implement advanced ML models tailored for time series analysis, i.e., Long Short-Term Memory-based (LSTM) networks (Hochreiter and Schmidhuber, 1997). We follow different model configurations to
exploit the sequential nature of the training data: we use unidirectional and bidirectional learning mechanisms. The first mechanism analyzes the input data in a sequence from the first time step to the last, whereas the latter mechanism additionally considers the reversed sequence from the last time step to the first, respectively. Moreover, to explicitly enable spatiotemporal modeling, i.e., encode topological and spatial contextual relationships, we also implement ConvLSTM models (Shi et al., 2015). Consequently, in the experimental evaluation, we exhaustively disentangle the prediction accuracies as a function of
the actual prediction model, the learning mechanism, and the deployed driving factors, i.e., different feature sets used for the prediction, respectively. Experimental results are obtained from Peru's capital Lima and Callao, which features a high population dynamic. To gain insights regarding the competitive performance of LSTM-based models in this application context, we also deploy multilinear regression (MLR) and RF models for comparison.

Regarding the application context of this study, solely a few works explicitly focused on applying time series ML methods for
mapping future natural hazard-related exposure and vulnerability. For instance, Johnson et al. (2021) simulated future urban land use changes up to the year 2050 with a trend-based logistic regression cellular automata model and evaluate potential flood exposure for the Philippines. Scheuer et al. (2021) model residential-choice behavior on a city level and examine how this process can translate into future trends regarding exposure, vulnerability, and risk. Here, *from an epistemological point of view*, we uniquely combine the forecasted population data with earthquake and tsunami hazard models to quantify the future
number of people living in earthquake-exposed and tsunami-exposed areas in Lima and Callao, Peru.

The remainder of the paper is organized as follows. In Sect. 2 we detail the proposed methodology. We describe the study area and experimental setup in Sect. 3. Experimental results are revealed in Sect. 4 and concluding remarks are given in Sect. 5.



## 2 Material and Methods

Figure 1 provides an overview of the proposed workflow for spatiotemporal forecasting of population data and quantification
of exposure. First, multitemporal gridded population data is compiled and aligned to a set of geospatial covariates, i.e., driving
factors. The data is fed into the LSTM-based models to establish a population forecast. The modelled future population is
utilized with hazard models to quantify the number of people living in earthquake-exposed and tsunami-exposed areas in Lima
and Callao, Peru, in the year 2035.

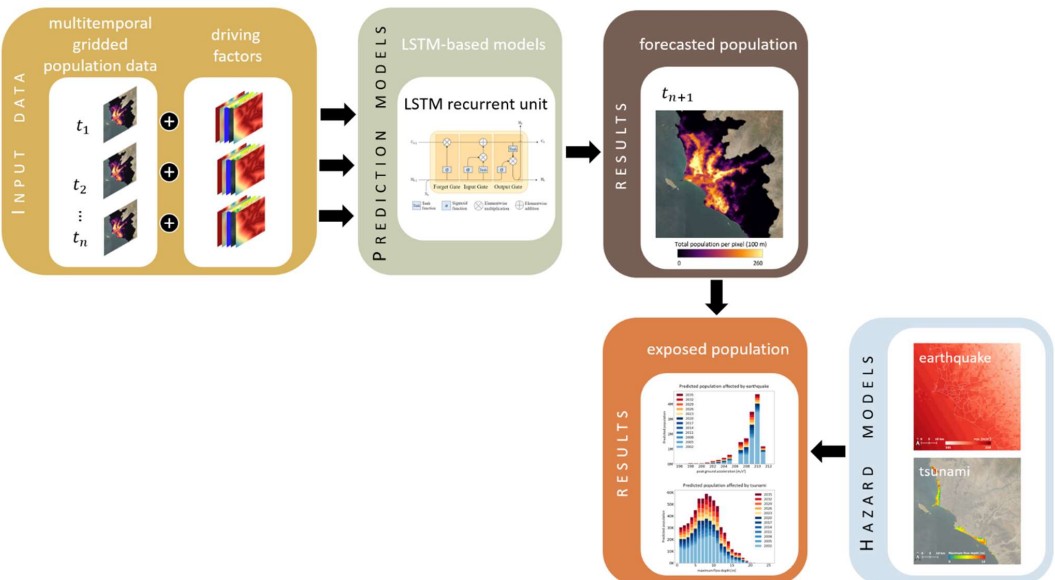

**Figure 1: General workflow for the spatiotemporal forecasting of population data in earthquake and tsunami-exposed areas of Lima and Callao, Peru.**

### 2.1 Multitemporal gridded population data

As the key input variable for the spatiotemporal forecasting of population, we harvest multitemporal gridded population data
from the WorldPop initiative (Lloyd et al., 2017; Stevens et al., 2015). The data set consists of annual multitemporal gridded
population data with a spatial resolution of 100 meters for the period 2000-2020, which describes the residential population
(Fig. 2). Thereby, WorldPop provides population counts which were adjusted to the United Nations population estimations
(United Nations, 2022). The data set was created with a regression-tree-based semi-automated dasymetric modeling approach.
First, a weighting layer was created with an RF approach and multiple spatial covariates, including country-specific census
counts, land cover, Digital Elevation Model (DEM), nighttime lights, net primary productivity, weather data, road networks,
waterbodies and waterways, protected areas, and 'facility' locations such as hospitals, schools etc. The modeled layer was
subsequently deployed to perform a dasymetric redistribution of the census counts at a country level (Stevens et al., 2015).
Actual census counts are redistributed from the smallest available administrative unit to the population grid with a higher
spatial resolution. The modeled layer determines the weight of the population for each grid cell. Figure 2 also displays the
absolute population change per grid cell for the time interval 2000-2020. It is traceable that the core of the settlement area
faced a decrease in population, while the vast majority of grid cells document an increase in population over the last two
decades.



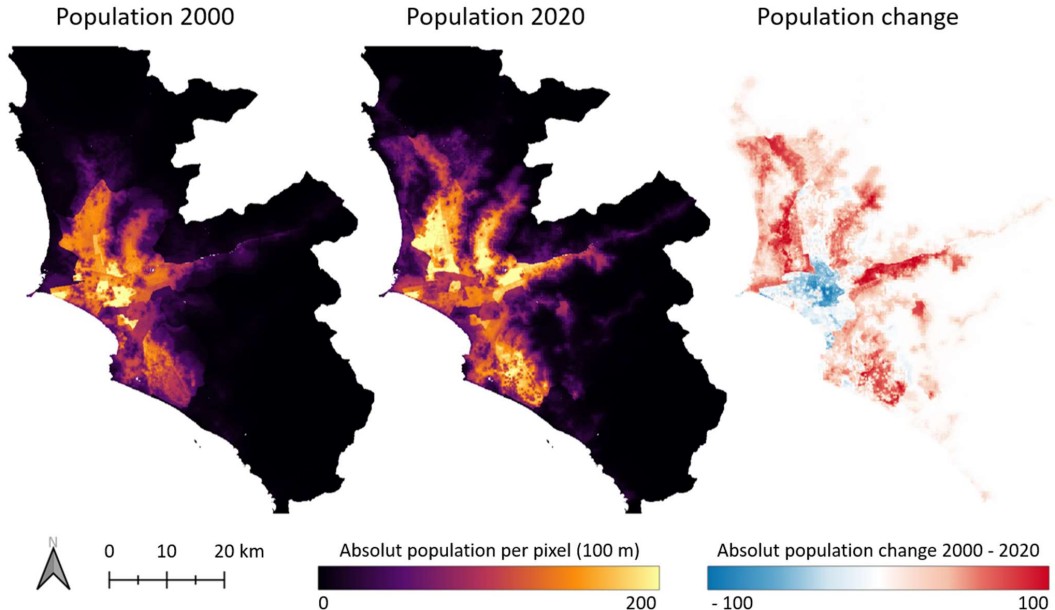

**Figure 2: Starting point (population of the year 2000) and end point (population of the year 2020) of the annual gridded population WorldPop time series data, which serves as input for the forecasting models, and corresponding visualized absolute population change between 2000-2020 for Lima and Callao, Peru.**


### 2.2 Driving factors

We compute a set of geospatial covariates, i.e., driving factors, for spatiotemporal forecasting of population data. The driving factors are either time-variant or time-invariant (Fig. 3). Time-variant driving factors vary substantially over time and must thus be computed consistently along the timely resolution of the time series data, whereas the latter remain rather static over

time. Land cover is an important driving factor for describing urban dynamics. The Moderate Resolution Imaging Spectroradiometer (MODIS) Land Cover data (Fig. 3a) from the National Aeronautics and Space Administration (NASA) has been provided annually since 2001 and thus matches the temporal resolution of the population data (Friedl and Sulla-Menashe, 2019). We group the thematic classes of the data set into four distinctive categories, i.e., "vegetation" "built-up", "barren", and "water". From this multi-class data set, one-hot layers were created for each of the four thematic classes to be used as input

for the models. Besides, the data feature a spatial resolution of 500 meters, which corresponds to the coarsest resolution among all used input features. Consequently, we compute the second time-variant driving factor, i.e., distance to the boundary of built-up areas (Fig. 3b) based on the Euclidean distance function, by deploying the higher spatially resolved multitemporal gridded population data sets (Sect. 2.1). Especially the natural conditions of an area shape geospatial change trajectories. One very important geographic input factor for population dynamics is the terrain since human settlements mostly appear on terrains

with flat or solely moderate slopes (Dobson et al., 2000). In this study, the Copernicus DEM (ESA, 2022), provided by the European Space Agency (ESA), with a spatial resolution of 30 meters was used to compute slope estimates (Fig. 3c). The Copernicus DEM data set also contains information about water bodies, which were combined with the water bodies contained in the OpenStreetMap (OSM) data set (2022) to compute a layer indicating the distance to water for the study area (Fig. 3d). The OSM data set also served for the compilation of geospatial vector data representing roads and computing distances thereof

(Fig. 3e). Lastly, we also computed a distance grid to the city center. In this study, the center of our study area was defined as a point coordinate situated between the current central business district and the historic city center, i.e., the Centro Histórico of Lima (Fig. 3f). The collected driving factors represent frequently adopted variables in studies of predicting geospatial change trajectories (Gómez et al. 2020; Liu et al. 2017; Pijanowski et al. 2002).





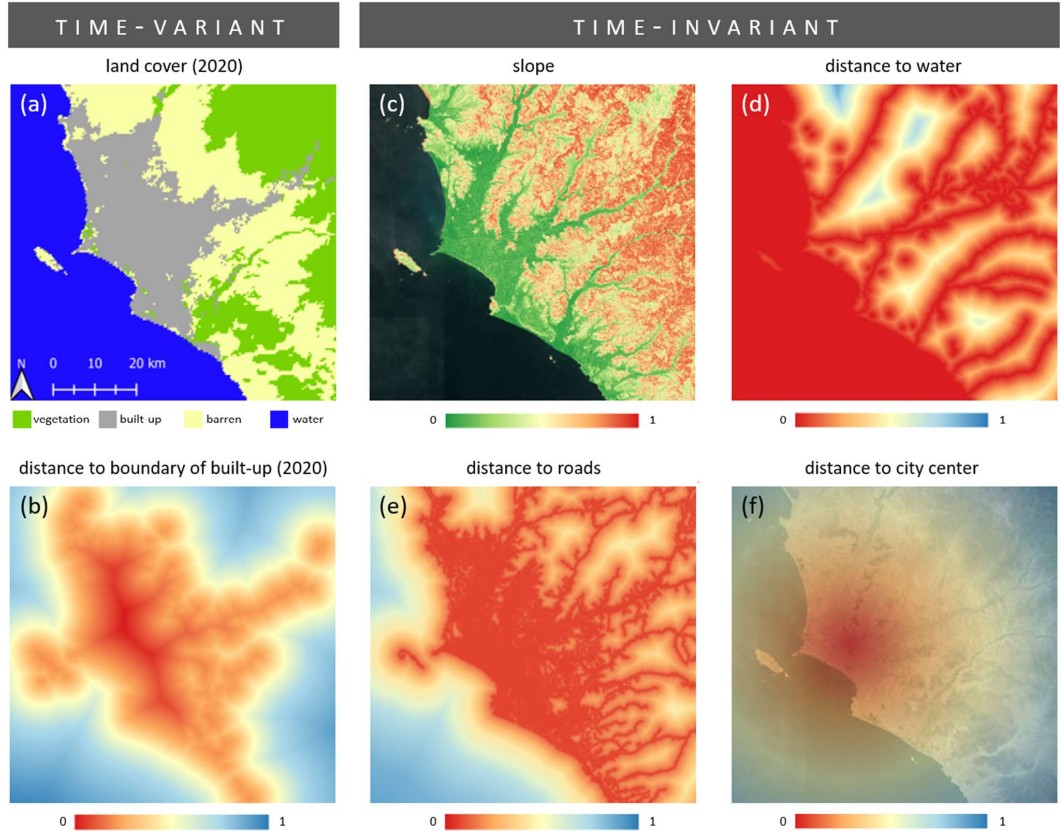

**Figure 3: Driving factors deployed for spatiotemporal forecasting of population data.**

**2.3 LSTM-based models**

The population data of time steps $t_1, t_2, \ldots, t_n$ and the corresponding driving factors are concatenated as the input for the LSTM models, which map the input to a prediction of the population at time step $t_{n+1}$. Generally, LSTM models belong to the family of recurrent neural networks (RNN). The latter represents a generalization of feedforward neural networks with internal memory and which are designed to process sequential information (Rumelhart et al., 1986). RNNs model the input as sequentially arranged time steps while preserving the information of each past element as state memory in the hidden unit (LeCun et al., 2015). Such networks are referred to as recurrent, since the architecture is repeated over the time steps, whereby the weights are shared in the different temporal layers and the underlying function remains fixed over all time steps (Aggarwal, 2018). However, Hochreiter and Schmidhuber (1997) introduced LSTM networks to overcome the problem of vanishing gradients. LSTM networks are equipped with complex blocks as hidden layers. Those blocks implement gates and memory cells, which control the flow of information and accumulate the state information in order to obtain the capability of long-term memory. The blocks, or the so-called 'LSTM-cells', contain an internal recurrence mechanism additional to the outer recurrence mechanism of the RNN (Goodfellow et al., 2016). Thus, LSTMs can be considered for sequence learning and forecasting, especially when long-term dependencies should be encoded from the input data. The main model architecture comprises an LSTM unit with the equations in (1):

$$i_t = \sigma(W_{xi}X_t + W_{hi}H_{t-1} + W_{ci} \circ C_{t-1} + b_i) \tag{1}$$

$$f_t = \sigma(W_{xf}X_t + W_{hf}H_{t-1} + W_{cf} \circ C_{t-1} + b_f)$$





$$C_t = f_t \circ C_{t-1} + i_t \circ tanh(W_{xc}X_t + W_{hc}X_{t-1} + b_c)$$

$$o_t = \sigma(W_{xo}X_t + W_{ho}H_{t-1} + W_{co} \circ C_t + b_o)$$

$H_t = o_t \circ \tanh(C_t)$

where $X_t$ represents the input to the cell, $C_t$ the memory state, and $H_t$ the hidden state. The notation '∘' denotes the Hadamard product or element-wise product. In the equations, $i_t$, $f_t$, and $o_t$ refer to the input, forget, and output gates, respectively, $t$ is the time-step, $\sigma$ the sigmoid activation function, tanh the hyperbolic tangent function, and $W$ are the weight matrices and $b$ the biases, respectively (Fig. 4a).

To enable spatiotemporal modeling, ConvLSTMs were employed. ConvLSTMs further contain convolutional structures with respect to both the input-to-state and state-to-state transitions. Thus, ConvLSTMs predict the future state of an entity (e.g., image pixel) from the current and past states of its surrounding entities (Shi et al., 2015). The inputs, cell outputs, hidden states, and gates are 3-dimensional tensors with rows and columns of the 2-dimensional input image as the last two dimensions. The internal operations thus use convolutions, which encode the spatial information (ibid.). The architecture of a ConvLSTM is

similar to an LSTM with the addition of the convolutional operator (Fig. 4b). Equations in (2) describe the ConvLSTM, which differ from the LSTM equations regarding the convolution operator, denoted by '*':

$$i_t = \sigma(W_{xi} * X_t + W_{hi} * H_{t-1} + W_{ci} \circ C_{t-1} + b_i) \tag{2}$$
$$f_t = \sigma(W_{xf} * X_t + W_{hf} * H_{t-1} + W_{cf} \circ C_{t-1} + b_f)$$
$$C_t = f_t \circ C_{t-1} + i_t \circ \tanh(W_{xc} * X_t + W_{hc} * X_{t-1} + b_c)$$

$o_t = \sigma(W_{xo} * X_t + W_{ho} * H_{t-1} + W_{co} \circ C_t + b_o)$
$$H_t = o_t \circ \tanh(C_t)$$

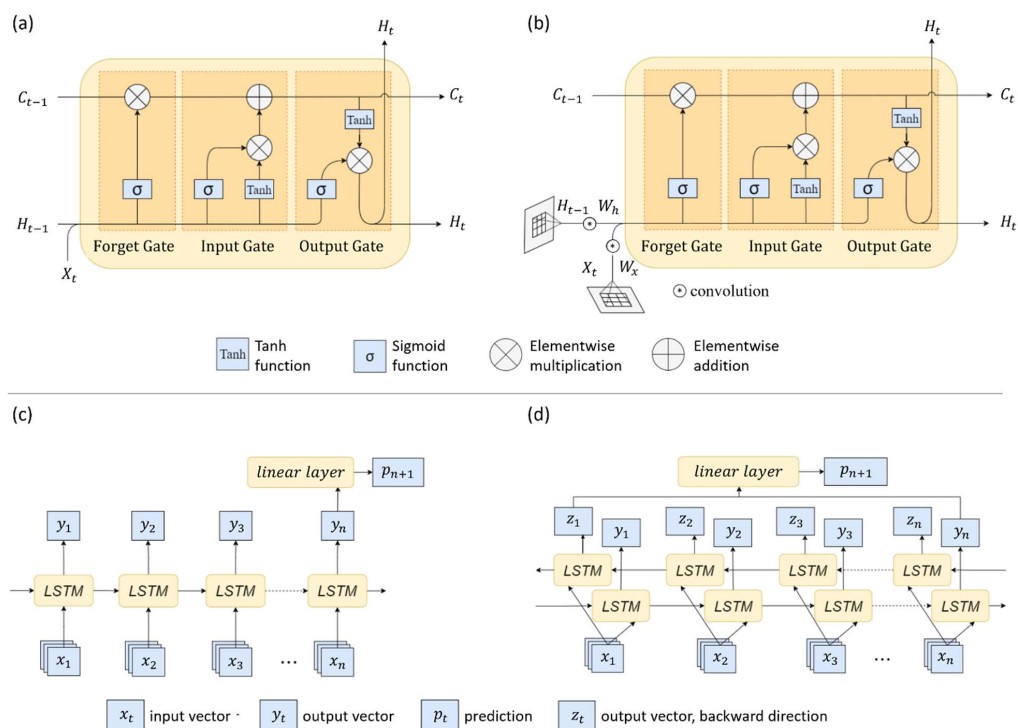

**Figure 4: Implemented LSTM components and network architectures: a) LSTM cell, b) ConvLSTM cell, c) unidirectional forward network, and d) bidirectional network.**




In this study, we train both models with a unidirectional forward (Fig. 4c) and bidirectional (Fig. 4d) learning mechanism, respectively. $x_t$ refers to the input data stacks for the chosen input years, i.e., $x_1, x_2, x_3, \ldots, x_n$. $y_t$ and $z_t$ are the generated output vectors at each time step, with the chosen hidden dimension of 64. For the bidirectional network, the forward layer outputs $y_t$ are computed iteratively using the input data in a sequence from the first time step to the last, while the backward layer outputs $z_t$ are calculated from a reversed sequence of the inputs from the last time step to the first. To retrieve a one-dimensional output that represents the predicted population number $p_{n+1}$ at time step $n + 1$, a linear layer is applied to the last output vectors. For the bidirectional networks the last outputs $y_n$ and $z_n$ were concatenated. The ConvLSTM networks have the same architecture as in Fig. 4 c-d, but with convolutional operations included in the ConvLSTM cells as shown in (Fig. 4b).

### 2.4 Hazard models: Earthquake and Tsunami

Earthquake and tsunami simulation data for this study were provided by the RIESGOS 2.0 project, which focuses on the creation of scenario-based multi-risk assessment in the Andes region (RIESGOS, 2022). The simulations are based on the historical earthquake of the year 1746 with an off-shore epicenter and a magnitude of 8.9 (Gomez-Zapata et al., 2021). To assess the population affected by this earthquake and correspondingly triggered tsunami, spatially distributed peak ground accelerations (Fig. 5a) and maximum flow depths (Fig. 5b) are used, respectively. The ground motion fields are generated based on ground motion prediction equations according to Montalva et al. (2017). The tsunami simulations (Androsov et al., 2023) are based on parameters proposed by Jimenez et al. (2013). The two data sets are provided with 1-kilometer and 10-meter spatial resolution, respectively, and were resampled to the spatial resolution of 100 meters of the population grid for the exposure analysis.

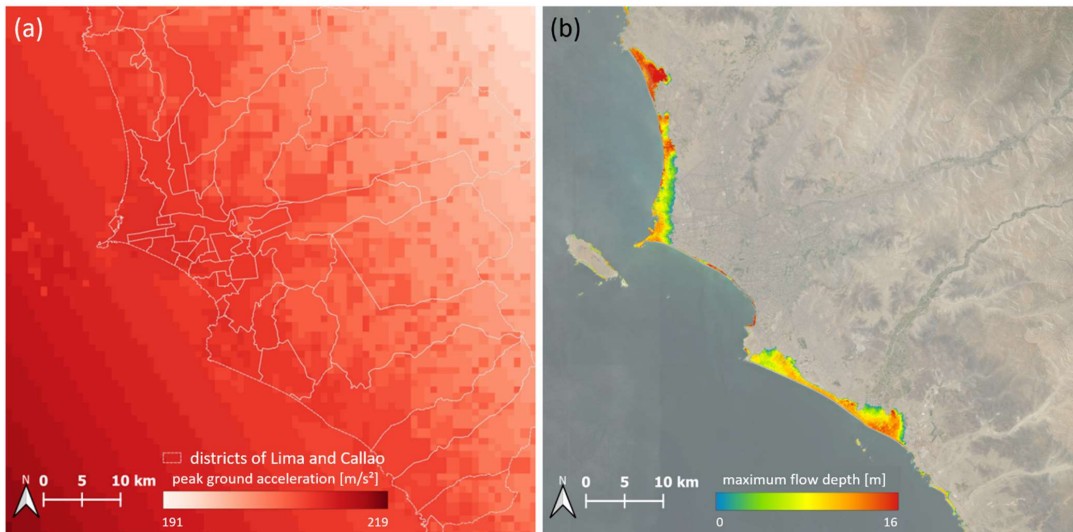

**Figure 5: Considered hazard models: (a) peak ground acceleration characterizing the historical earthquake of 1746 with an off-shore epicenter and a magnitude of 8.9 and (b) corresponding tsunami characterized by maximum flow depth.**

### 3 Experimental Setup

As aforementioned, the study area comprises the settlement area of Peru's capital Lima and the neighboring province of Callao, which has a spatial coverage of approximately 6500 square kilometers. All data sets were reprojected to the WorldPop projection EPSG:4326, the World Geodetic System 1984, and resampled to the spatial resolution of 100 meters of the gridded population data. All layers were normalized individually and then stacked to a multidimensional array of the shape (20, 10,



888, 888). Thereby, the first position carries the twenty years of gridded population data, whereas the second position contains the driving factors, while establishing an image size of 888 * 888 elements. The WorldPop time series data is deployed along

a three-year interval (which provides a suitable tradeoff here between the forecasting capability of the model and having a sufficient number of time steps available for training the model) and split into training data set and validation data set along the temporal dimension. The training data set contains the earlier six time steps (2002, 2005, …, 2017), whereas the validation data set contains the later six time steps (2005, 2008, …, 2020) of the time series. In both training and validation data sets, the variables of the first five time steps were adopted as input and the last time step was used as the ground truth labels. As such,

the target of the training data set is to predict the population of the year 2017, and the goal of the validation data set is to forecast the population map for the year 2020. (Fig. 6).

We carry out experiments with three sets of driving factors deployed for forecasting: i) all driving factors described in Sect. 2.2; ii) solely the time-invariant driving factors, i.e., slope, and distances to water bodies, roads, and the city center, respectively; iii) the population data only. Here it can be noted that the latter two reduced sets of variables enable the prediction

of multiple time steps in the future. When including also time-variant driving factors, i.e., land cover and distance to the boundary of built-up areas, solely one future time step can be predicted: a model learns the changes between a specific time interval and can thus predict the same time interval in the future. Equation (3) describes this relation, with $t_{n+1}$ being the forecasted year, $t_n$ the year of the last input, and i the interval size:

$$t_{n+1} = t_n + i \qquad\qquad (3)$$

Here, we trained all models with an interval $i$ of three years including every third year as training data from the WorldPop time series data. To be more specific, if we input the years 2008, 2011, 2014, 2017, and 2020, the model is expected to produce an estimation for the year 2023. Subsequently, the year 2026 can be forecasted with the data of the year 2023 as input. The time-invariant driving factors can be assumed to be valid for 2023 too, and the corresponding population data was predicted prior. In contrast, no valid estimates for 2023 are available regarding the time-variant driving factors. Thus, excluding the time-

variant driving factors from the training enables iterative predictions of multiple intervals. This sliding time window approach was conducted similarly to the forecasting strategy deployed by Wang and Lee (2021) and is displayed in Fig. 6.

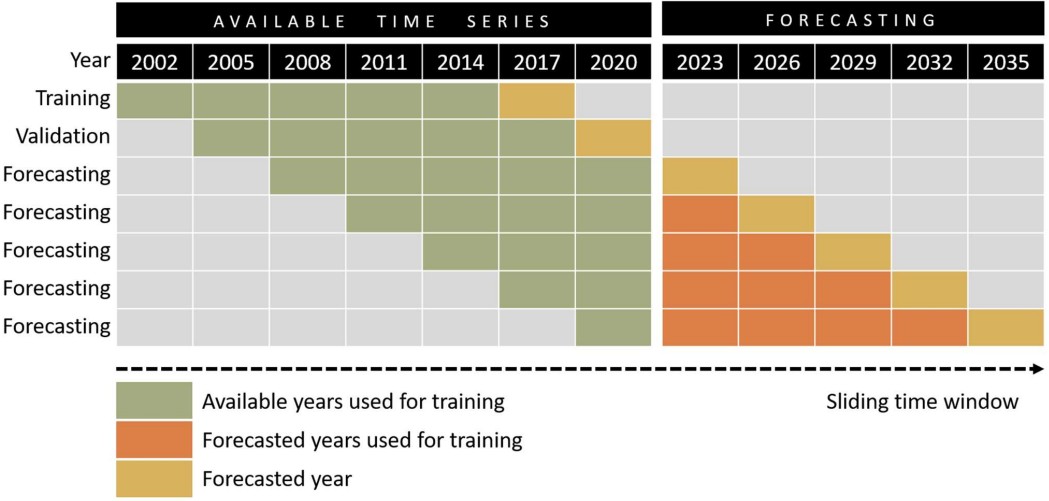

**Figure 6: Training, validation and forecasting concept: The training data set utilizes the earlier six time steps (2002, 2005, …, 2017), whereas the validation data set utilizes the later six time steps (2005, 2008, …, 2020) of the time series. The variables of the first five time steps were adopted as input and the last time step was used as the ground truth labels. As such, the aim of the training data set is to predict the population of the year 2017, and the aim of the validation data set is to forecast the population map for the year 2020, respectively. Forecasting beyond the year 2023 is obtained with a sliding time window strategy, where previous forecasted years are deployed for model training (adapted from Wang and Lee, 2021).**



All the tested models were trained for 50 epochs, the optimizer was Adam, the loss function was mean squared error loss, and

the initial learning rate was set to 0.0012 and was reduced by the factor 0.1 through a learning rate scheduler, when the error

reached a minimum plateau. To evaluate the proposed framework, two baseline methods were adopted, i.e., MLR and RF.

Thereby, the hyperparameters of RF were tuned heuristically as follows: ntree = 500 and mtry = 1,2, ⋯,51.

## 4 Experimental Results and Discussion

### 4.1 Model evaluation

To provide a first comparative overview regarding prediction accuracy, Fig. 7 contains scatter plots of the different methods

for the predicted year 2020. Thus, it illustrates the deviations of the forecasts (y-axis) concerning the actual population values

(x-axis). Each point corresponds to one grid cell in the study area. The color coding reflects the point density in %, the red line

is the regression line of forecasts and population values, and the black line corresponds to the identity line where $y = x$,

respectively. It reveals that all models feature a substantial concentration of the density along the identity line, which underlines

the overall validity of our setup. However, traceable differences regarding the different models exist. The majority of the

baseline models based on linear regression and RF feature a lower point density along the identity line compared to the LSTM-

based models, especially for grid cells with medium and high population values. The corresponding root-mean-squared error

(RMSE) values also clearly indicate that the LSTM-based models outperform both the ConvLSTM models and the baseline

methods. In detail, the uncertainty in terms of RMSE could be reduced from 4.298 (RF), 4.109 (MLR), and 3.946 (ConvLSTM

- bidirectional), respectively, to 3.629 (LSTM - bidirectional) while maintaining an excellent model fit ($R^2$ = 0.995). Along

this line of models, this corresponds to an increase of more than 14 percentage points in terms of model accuracy.

It can be noted that using the static features for the baseline models, i.e., MLR and RF, and solely deploying the population

data for the LSTM and ConvLSTM with bidirectional learning mechanism enabled the respective best predictions.

Counterintuitively, the LSTM models outperform the ConvLSTM models unambiguously. Past works showed that the

inclusion of additional spatial context information via ConvLSTMs can be beneficial for increasing prediction accuracy (Shi

et al., 2015; Gavahi et al., 2021). However, in our idiosyncratic data setting, some inconsistencies in the WorldPop data can

be found: water bodies, conservation areas, or industry districts were traceably not masked during the disaggregation, which

lead to mostly non-zero grid cell values in these areas. Solely individual grid cells lying in these regions hold zero values in

the WorldPop data. All convolutional models predict these grid cells with non-zero population, as they learn from the

surrounding grid cells. This can be seen in Fig. 7 at $x = 0$, where the actual population is zero, but the prediction differs quite

strongly. Nevertheless, across all models, MAE values indicate a deviation of less than three people per grid cell, which stresses

the overall soundness of our setup.



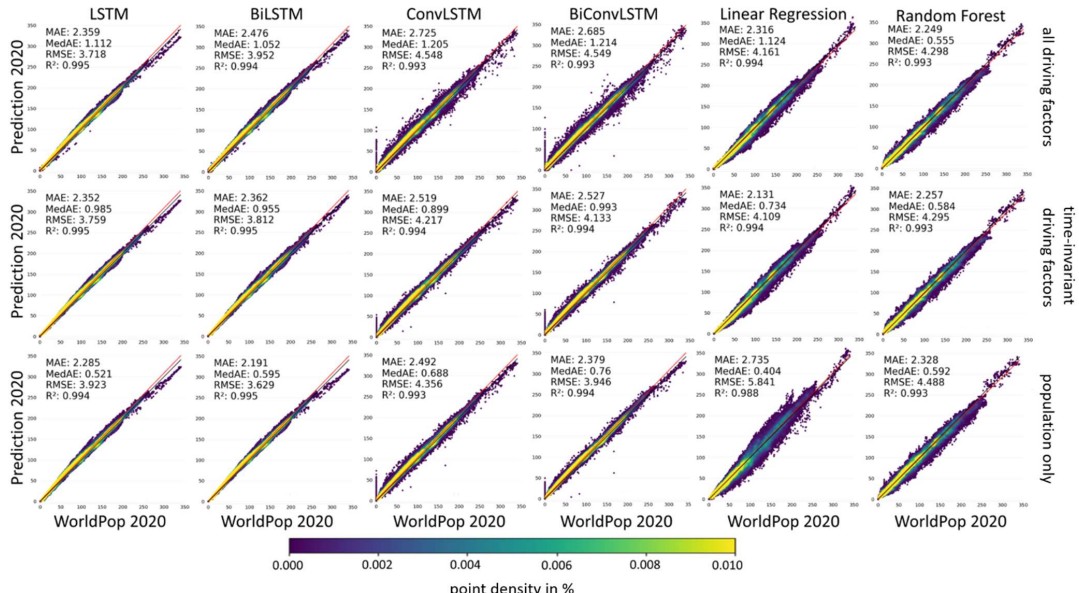

**Figure 7: Scatter plots and corresponding error measures, i.e., mean absolute error (MAE), median absolute error (MedAE), RMSE, and R², respectively, for the predicted year 2020 as a function of the actual prediction model, the learning mechanism, and the deployed driving factors.**

Figure 8 provides prediction differences to the actual numbers of 2020 from a spatial perspective. Grid cells with overestimated population numbers are colored in green, whereas grid cells with underestimated population numbers are colored in red. Thereby, it can be traced that the LSTM-based and ConvLSTM-based predictions overestimate population numbers for the majority of grid cells, while both the MLR-based and RF-based predictions underestimate population numbers for the majority of grid cells (also revealed by the regression line in Fig. 7). However, both the LSTM-based models and ConvLSTM-based models follow consistently the overall trend of an area: in tendency, they exaggerate population numbers in areas of an increasing population and underestimate population numbers in areas of a decreasing population (see also Fig. 2 for a visualization of areas of increasing and decreasing population numbers in Lima and Callao). The baseline models do not reflect this overall trend, whereby over- and underestimations are more dispersedly distributed across the study area.





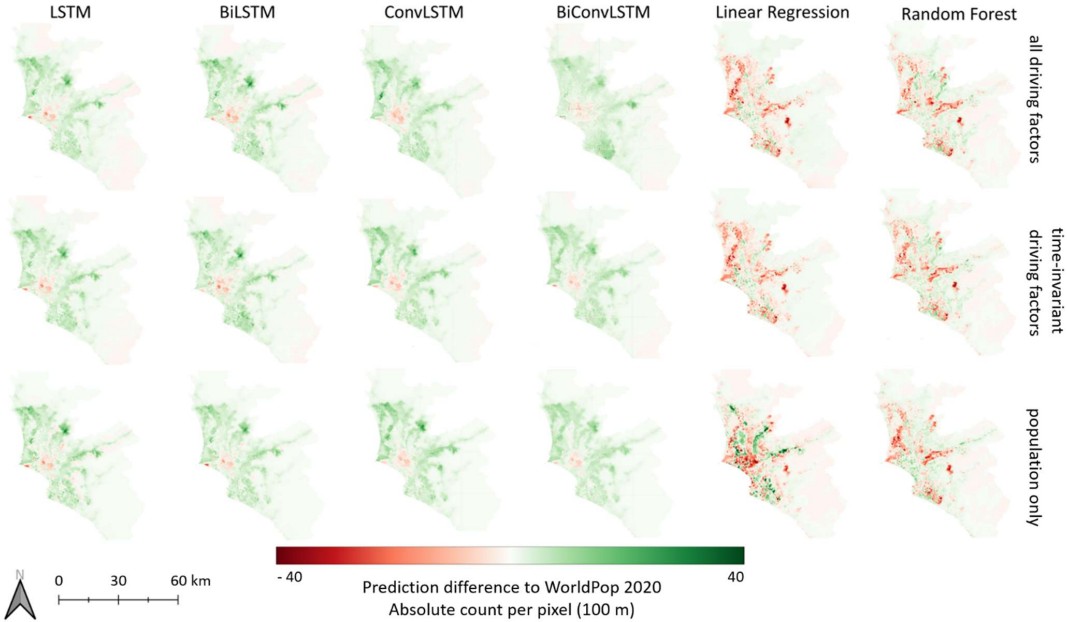

**Figure 8: Maps of prediction differences of the models with respect to the actual numbers of 2020.**

### 4.2 Population forecasting

The actual population forecasting, which is deployed for the subsequent exposure analysis, was carried out based on the most

favorable model, i.e., the LSTM learned on the population data with a bidirectional learning mechanism. We implemented this
model in our forecasting concept (Fig. 6), where forecasting beyond the year 2023 is obtained with a sliding time window
strategy, i.e., previously forecasted years are deployed for model training. Figure 9 displays both the forecasted population and
the change between subsequent time steps until the year 2035. Thereby, the population increases by about 3.6 million, which
accounts for 35 % of Lima's population in 2020.




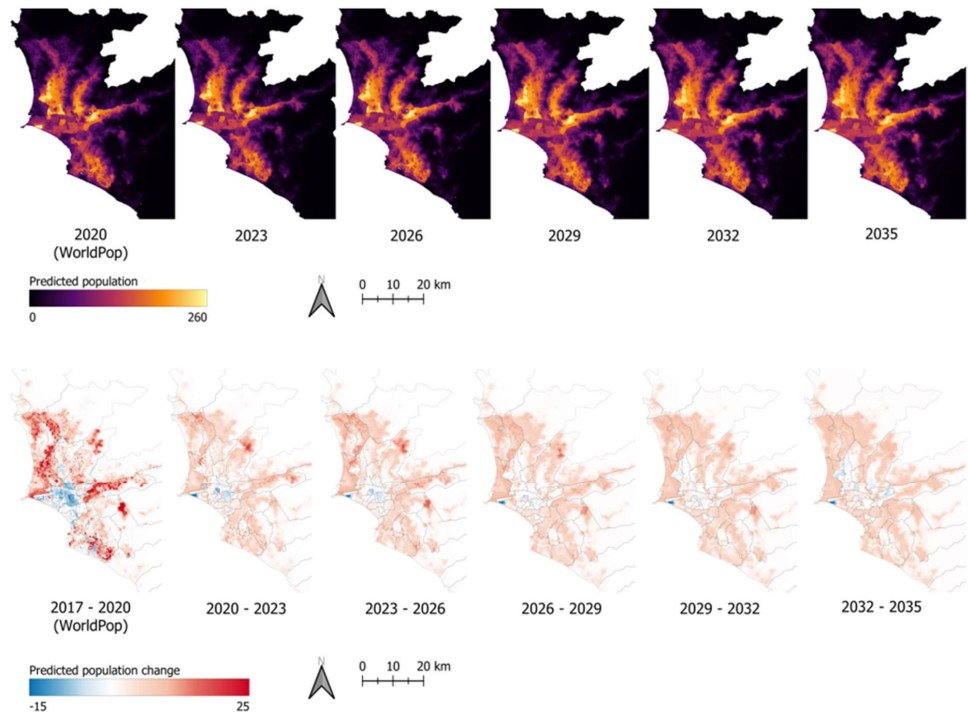


**Figure 9: The upper part provides a visualization of the WorldPop data for the year 2020 and forecasted population until the year 2035 along a three-year interval. The lower part contains the corresponding predicted change of the population for the different time intervals.**

### 4.3 Exposed future population

The hazard models (Sect. 2.4) and the predicted population distribution (Sect. 4.2) are deployed to compute the future population count as a function of different hazard intensity levels. Figure 10 provides accumulated population numbers for different levels of peak ground acceleration (a) and maximum flow depth (b) along the 3-year time interval. It can be observed that the majority of future population, i.e., 12.5 million inhabitants, lives in areas of a high peak ground acceleration, i.e., PGA $\geq 207$ m/s². These numbers of future exposed population are induced by a growth of almost 30 % over the forecasted time

span. This is less than the 35 % growth in the whole Lima Metropolitan area. However, already today more than 82 % of Lima's inhabitants reside in these districts. Consequently, this growth accounts for more than 70 % of the predicted additional people in Lima. Furthermore, more than 600,000 people are anticipated to live in areas which are at peril of a maximum tsunami flow depth of more than two meters. The population in the tsunami inundation area will grow by 61 % until 2035, much more than the average growth of 35 %. Waves of up to 20 meters are modeled, and most of the affected people would

be hit by waves of from 6 to 11 meters. The areas with the largest modeled waves of more than 12 meters only have a small part of the population today, but these will even double in the forecasted 15 years.



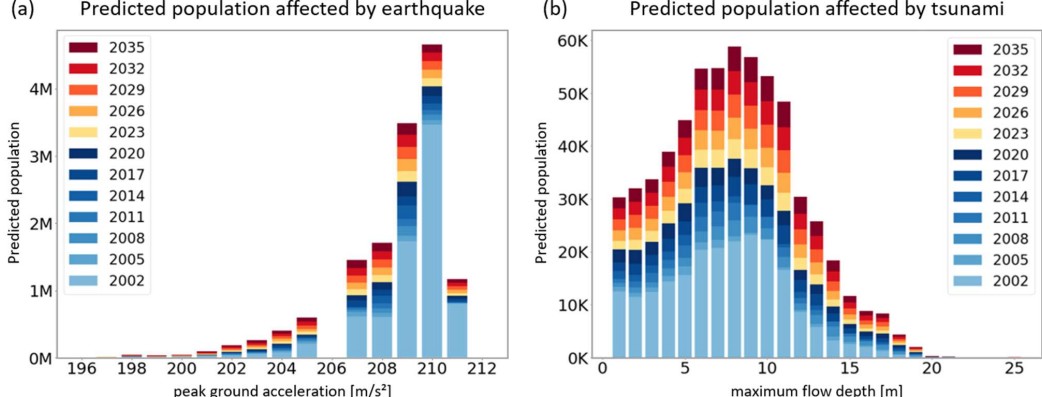

**Figure 10: Predicted number of people affected by different hazard intensity levels along a 3-year interval (2002-2035) regarding earthquake (a) and tsunami (b).**

The forecasted spatial distribution of the population along with hazard intensities is visualized in Fig. 11 from a south-western viewing angle. We aggregated the grid cells from the 100-meter resolution to a 1-kilometer resolution for visual representation. The visual inspection uncovers new future hot spots of the exposed population, i.e., areas that face simultaneously high population increases and severe hazard intensities, such as Lima and Callao's north-western and south-western settlement areas along the coastline. Anticipating those patterns can help urban planners and policy makers to proactively develop

effective strategies for risk mitigation.





**Figure 11: Maps of the predicted population affected by earthquake (upper figure) and tsunami (lower figure) for the year 2035 with corresponding hazard intensities.**




## 5 Conclusions and Outlook

In this paper, we learned population-related geospatial change trajectories over time and provide population forecasts for Peru's capital Lima and Callao to identify future hotspots of earthquake and tsunami exposure. The experimental results underline the superior performance of temporal models, i.e., LSTM-based networks, in accurate forecasting of the changes in

distribution. Given that the source dataset of the tested data is openly accessible and has global coverage, our workflow can be generalized to forecast population changes in other locations with solely a few adaptations (e.g., determine the best model hyperparameters empirically for a specific area/data set) for optimal forecasting accuracies.

Several extensions can be explored in future work: foremost it is crucial to obtain a picture on future risks and not solely on aspects of the exposure, i.e., the population at risk. This would require the collection of time series data for model training

with multiple risk-related target variables including population, building types, occupancies, among others, to also align vulnerability information, i.e., earthquake and tsunami-related fragility functions. Thus, a more thorough forecasting of future earthquake and tsunami risks can be conducted. From a methodological point of view, the consideration of multiple risk-related target variables also enables the development of multitask learning models, which can encode interdependencies between the considered target variables to enhance the prediction accuracy (Geiß et al., 2022). Also, a multitask model is able to learn the

time-variant driving factors for enhanced forecasts and, thus, drawing a more robust picture of future risks.

### Author contributions

All authors contributed to the idea and scope of the paper. CG, JM, ESO, and YZ contributed to conceptualization, data curation, methodology, and software, SH and JCGZ computed and integrated hazard models, and ES, CG, SH, and JCGZ were responsible for funding acquisition and ES for project management. CG, JM, and ES prepared the initial manuscript, which

was reviewed and edited by the co-authors. All authors have read and approved the final version of the manuscript.

### Competing interests

ES is guest editor of the journal. The other authors declare that they have not conflict of interest. The funders had no role in project design, data collection and analysis, decision to publish, or preparation of the manuscript.

### Financial support

This study has been partially funded by the German Federal Ministry of Education and Research (BMBF) within the project RIESGOS 2.0 (03G0905A-C).

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
