# Peer review of "Anticipating a risky future: LSTM models for spatiotemporal extrapolation of population data in areas prone to earthquakes and tsunamis in Lima, Peru"

_EGUsphere, 2023_

## Author Response (AR1)

**Anticipating a risky future: LSTM models for spatiotemporal extrapolation of population data in areas prone to earthquakes and tsunamis in Lima, Peru**

**November 2023**

Christian Geiß, Jana Maier, Emily So, Elisabeth Schoepfer, Sven Harig, Juan Camilo Gómez Zapata and Yue Zhu

We would like to thank the reviewer very much for the very valuable and helpful comments. We considered the remaining point and carefully revised the manuscript accordingly. Please find our detailed response to the comment below.

In the response below, the comment of the reviewer is reported in *italic*. Responses from the authors do not feature any special formatting. Quotations from the manuscript in its revised or in its original form are reported in blue.

**Reviewer #1:**

*The manuscript represents a valuable contribution to the prediction of future hotspots of earthquake and tsunami exposure.*

*The title and abstract are relevant, succinctly encapsulating the paper's content. The introduction boasts a commendable array of references, which could be further enhanced in terms of quantity and relevance. The Materials and Methods section is quite comprehensive, while the Results and Discussion are well-structured, complemented by various supportive figures. The conclusions are succinct but clear.*

We thank the reviewer for his/her generally positive judgement and genuinely appreciate the suggestions for improvement.

*Additionally, I would like to suggest the following improvements:*

*The abstract lacks essential data pertinent to the conducted analysis, specifically information regarding the precision of the methods employed and the reliability of the results.*

Thank you for sharing this observation. We provide more quantitative number regarding the model accuracies as follows now:

The results clearly underline the value of the LSTM-based models for forecasting gridded population data: The most accurate prediction obtained with an LSTM equipped with a bidirectional learning scheme features a root-mean-squared error of 3.63 people per 100 *100 meters grid cell, while maintaining an excellent model fit ($R^2 = 0.995$). We deploy this model for anticipation of population along a three-year interval until the year 2035.

*Is Figure 2 too large? Nevertheless, the text at the top can be resized to ensure compatibility with the main text.*

Thank you for this observation. We resized the text accordingly and provide a revised figure in the new manuscript.

*For Figure 5a's legend, it would be advisable to ensure that the text does not overlap with the image territory.*

Thank you for sharing this observation and suggestion. We changed the legend in a way that the legend does not overlap anymore with relevant image content. According to our perspective, that was the case for district boundaries that were partially affected by the placement of the legend – this is not the case anymore.

*On page 8, row 217: "Is the equation in (3) the same as before?"*

We modified the sentence and hope that it is less ambiguous now:

Equation (3) describes this relation:

$$t_{n+1} = t_n + i \tag{3}$$

with $t_{n+1}$ being the forecasted year, $t_n$ the year of the last input, and $i$ the interval size.

*Regarding Figure 6, the caption appears somewhat lengthy. Have you considered incorporating some of the description into the main text?*

Thank you very much for this observation. We shortened the caption but still aimed to keep it somewhat self-contained, since it could be difficult to understand the rather complex content from the text only.

*Figure 8's clarity could be improved as the details seem somewhat indistinct; the choice of subtle colors may have influenced the outcome.*

Thank you for sharing this point. We agree that the LSTM-based and ConvLSTM-based outcomes feature a pattern that is not prominently distinct in this plot. However, the overall goal is to particularity highlight differences w.r.t. the MLR and RF models. We hopefully make this clear by relating to this aspect in the corresponding text:

Figure 8 provides prediction differences to the actual numbers of 2020 from a spatial perspective. Grid cells with overestimated population numbers are colored in green, whereas grid cells with underestimated population numbers are colored in red. Thereby, it can be traced that the LSTM-based and ConvLSTM-based predictions overestimate population numbers for the majority of grid cells, while both the MLR-based and RF-based predictions underestimate population numbers for the majority of grid cells (also revealed by the regression line in Fig. 7). However, both the LSTM-based models and ConvLSTM-based models follow consistently the overall trend of an area: in tendency, they exaggerate population numbers in areas of an increasing population and underestimate population numbers in areas of a decreasing population (see also Fig. 2 for a visualization of areas of increasing and decreasing population numbers in Lima and Callao). MLR-based and RF-based predictions do not reflect this overall trend, whereby over- and underestimations are more dispersedly distributed across the study area.

*Concerning Figure 11, I'm unclear about the significance of the "pop increase 2020 - 2035" bar. Could you provide clarification on its meaning?*

Thank you very much for identifying this unclear point. We substantially extended the caption to hopefully communicate the content of the figure better:

Maps of the predicted population affected by earthquake (upper figure) and tsunami (lower figure) for the year 2035 with corresponding hazard intensities. The solid grey bars indicate the population of the year 2020. The additional colored bar (on top) or textured bar indicate the estimated increase or decrease of the population until the year 2035, respectively. The corresponding color coding indicates the hazard intensity.

**Reviewer #2:**

*This is a very good paper that examines a critical topic for disaster risk reduction (DRR): the estimation of the actual populations in areas exposed to hazards and, moreover, the prediction of how these populations might change within a given time threshold. Commonly, this information is either unavailable or unupdated (especially in countries belonging to the Global South), so this paper makes a significant contribution along these lines.*

*The manuscript is clearly arranged and written, including an introduction section that delivers a proper literature review and describes the study's rationale and objectives and a materials and methods section that explains the sources and methodologies applied in this research. The results are presented and discussed with proper clarity and depth (and the included figures give sufficient support for this). It follows my suggestions for further improving the manuscript:*

We thank the reviewer for his/her generally positive judgement and genuinely appreciate the subsequent suggestions for improvement.

*First, I think that the authors should deliver a more profound argument about the choice or identification of what they call driving factors ("different feature sets used for the prediction", "geospatial covariates"). They point out that "the collected driving factors represent frequently adopted variables in studies of predicting geospatial change trajectories (Gómez et al. 2020; Liu et al. 2017; Pijanowski et al. 2002)" (page 4, line 132), but given the critical importance of the specific choice of factors, I think that more explanation or rationale should be provided.*

Thank you very much for discussing this point with us. We added additional content on the issue and also justified the selection of variables better, i.e., justify that we cover the main variable categories:

The compilation of a powerful and exhaustive set of geospatial covariates is frequently a challenge in the context of implementing a robust study design. For instance, Zhu (2023) lists more than 50 predictor variables which were employed in existing studies of land use and land cover prediction. Here, the collected driving factors represent frequently adopted variables in studies of predicting geospatial change trajectories (Gómez et al. 2020; Liu et al. 2017; Pijanowski et al. 2002). In detail, we internalize the main variable categories (Zhu, 2023), i.e., land use-related variables (Fig. 3a-b,f), environmental variables (Fig. 3c,d), infrastructural variables (Fig. 3e), as well as socio-economic variables (Fig. 2).

*Second, I believe the implications of the findings in section 4.3 require further discussion. The paper delivers an argument in terms of identifying the exposure of populations to tsunami and earthquake hazards, but after that, the manuscript only points out that "Anticipating those patterns can help urban planners and policy makers to proactively develop effective strategies for risk mitigation" (page 13, line 309). For instance, considering a certain degree of uncertainty, how could these patterns be integrated into planning? How might they lead to a change in DRR policies, considering the possibility of building and assessing different development scenarios? I know these objectives are beyond the scope of the paper, but enhancing this section would improve the paper's multidisciplinary interest. Reviewing additional literature could help to do this.*

Thank you very much for identifying those points and the suggestions. Consequently, we extended the end of section 4.3 and hopefully address the raised points:

Anticipating those patterns can help urban planners and policy makers to proactively develop effective strategies for risk mitigation. For instance, the created information about exposed population can be part of modern decentralized information systems for (multi)-risk assessment (Schöpfer et al., 2023). Here, one core element is to enable end users to explore various scenarios ("stories") of multiple hazards, cascading effects and their impacts by quantifying different *what-if* scenarios. Utilizing such a narrative-driven methodology empowers individuals to replicate diverse situations within a predetermined, multi-risk context, enabling them to assess and contrast outcomes. This multi-scenario approach proves invaluable for crafting strategies that fortify or enhance resilience, evaluating the effectiveness of proposed or already executed measures (e.g., benchmark scenarios) in the face of various hazard scenarios (acting as a 'stress test'), or in response to evolving conditions. Thereby, the importance of implementing mechanisms to visualize epistemic and aleatory uncertainties of the risk assessment procedure in graphical form is stressed to allow appropriate communication with end users.

*Some minor issues:*

*The concept of "natural disasters" (page 1, line 29) is not typically used in current literature about disaster risk reduction. I suggest using "socio-natural disasters" or similar.*

Thank you very much for this suggestion. We altered the paper accordingly and use the term "socio-natural disasters" consistently (used two times throughout the manuscript).

*In page 15, line 320, I think the word "population" is missing before "distribution".*

Thank you, adjusted accordingly.

*Throughout the manuscript, the authors use indistinctively passive and active verb forms ("were adopted", "we carry out", "our setup", "was carried out"). Please check for consistency.*

Thank you for this observation. We substituted passive wording with active sentences and aimed to consistently use past tense if applicable (we changed around 30 sentences).

---

## Author Response (AR2)

**Anticipating a risky future: LSTM models for spatiotemporal extrapolation of population data in areas prone to earthquakes and tsunamis in Lima, Peru**

**January 2024**

Christian Geiß, Jana Maier, Emily So, Elisabeth Schoepfer, Sven Harig, Juan Camilo Gómez Zapata and Yue Zhu

We would like to thank the editor very much for the very valuable and helpful comments. We considered the remaining point and carefully revised the manuscript accordingly. Please find our detailed response to the comment below.

In the response below, the comment of the editor is reported in *italic*. Responses from the authors do not feature any special formatting. Quotations from the manuscript in its revised or in its original form are reported in blue.

**Editor:**

In particular, please revise the following:

*1. It seems to me that there is a unit error in the Peak Ground Acceleration (PGA) values reported throughout the paper. PGA of the order of 200 m/s2 are ~ 20 g which is unphysical. I suppose it should read cm/s2 , please revise (see also Pulido et al., 2015: https://pubs.geoscienceworld.org/ssa/bssa/article/105/1/368/323556/Scenario-Source-Models-and-Strong-Ground-Motion)*

Thank you very much for identifying this glitch! We revised the whole paper accordingly using [cm/s²] as physical unit and also adapted Figure 1, 5, 10, 11.

*2. Please check the references to figures in the text. Sometimes you use "Fig." and other times you spell "Figure".*

Thank you for sharing this observation. We consistently use the abbreviation "Fig." across the text now.

*3. Please check the use of passive and active verbs (e.g. "were adopted", "was carried out", "we carry out") since they are used indistinctively. For instance, what it is done in the work, should be in present (e.g. we carry out, we adopt, we compute).*

Thank you very much for sharing this observation. We revised the whole paper accordingly and avoid the use of passive sentences generally and in particular when we refer to what was done in the work.

*4. P. 1, L34-35. "thus, induces the need to constantly update and accurately anticipate future geospatial population distributions in hazard-prone areas." Please revise the sentence, I think "thus" is misleading here. A possible alternative: "demands a frequent update and anticipation….."*

Thank you for this suggestion. We revised the sentence according to the suggestion.

*5. P.2, L41: "methods can be considered generally". Would it be "methods can be generally considered"?. Please check.*

Thank you for this suggestion. We revised the sentence according to the suggestion.

*6. P.2, L47: "However, especially recently, multiple authors". Please check the sentence, maybe "A second family of …. has been proposed recently".*

Thank you for this suggestion. We revised the sentence according to the suggestion.

*7. P.2, L56: "earth observation data are very valuable". Please revise the sentence. It may be something like "Earth observation is customarily used to…"*

Thank you for this suggestion. We revised the sentence according to the suggestion.

*8. P.2, L63-64: "from a data-oriented perspective, we mitigate the often expensive process of compiling time series data through innovatively make use of existing time series global population data sets". It is a cumbersome sentence, please check.*

Thank you for this suggestion. We revised and shortened the sentence accordingly.

*9. P. 2, L71: "encode topological and spatial contextual relationships". I think it should read "to encode ….". Please check.*

Thank you for this suggestion. We revised the sentence according to the suggestion.

*10. P.4, L123: "i.e., "vegetation" "built-up",". There is a comma missing between "vegetation" and "built-up". Please check.*

Thank you for identifying this error. We revised the paper accordingly.

*11. P.4, L128: "Especially the natural conditions of an area shape geospatial change trajectories.". Please check, this sentence is not clear to me.*

Thank you for this comment. We deleted the mentioned sentence and added content to the subsequent sentence: One very important geographic input factor for modelling population dynamics is the terrain, since human settlements mostly appear on terrains with flat or solely moderate slopes (Dobson et al., 2000).

*12. P.4, L129: "for population dynamics is the terrain". Specify that it is the "terrain topography", isn't it? Please check.*

Thank you for this suggestion. We revised the paper accordingly: one very important geographic input factor for modelling population dynamics is the topography of the terrain…

*13. P.4, L133: "indicating the distance to water for the study area". It might be more precise to write "water bodies". Please revise.*

Thank you for this suggestion. We revised the paper accordingly:

*14. P5, L137: "The compilation of a powerful and exhaustive set of geospatial covariates". It is not clear to me the meaning of "powerful and exhaustive" in this sentence. Should it read "large" dataset? Please check.*

Thank you for sharing this observation. We revised the paper as follows: The compilation of a set of geospatial covariates that enables accurate estimations is a frequent challenge.

*15. P5, L140: "variables in studies of predicting geospatial change trajectories". Should the sentence read "for predicting geospatial..."? Please check.*

Thank you for this comment. We deleted this unclear statement and write now: Here, the collected driving factors represent frequently adopted variables in past studies (Gómez et al. 2020; Liu et al. 2017; Pijanowski et al. 2002).

*16. P.5, Caption of Figure 3. "deployed for spatiotemporal forecasting of population data". Should it read "deployed for the spatiotemporal"? Please check.*

Thank you identifying this error. We revised the paper accordingly.

*17. P.6, L173. "The internal operations thus use convolutions". Is think the use of "thus" should be avoided here. Please check.*

Thank you for this comment. We deleted "thus" here.

*18. P8, Figure 5: Please check the units of the PGA.*

Thank you very much for identifying this glitch! We revised the whole paper accordingly using [cm/s²] as physical unit and also adapted Figure 1, 5, 10, 11.

*19. P8, L213: "which provides a suitable tradeoff here between". It is not clear for me the use of the word "suitable". Is it an acceptable tradeoff to have the smallest possible temporal windows? Please check.*

Thank you identifying this unclear point. We revised the paper accordingly: (which provides an acceptable tradeoff here between the forecasting capability of the model and having a sufficient number of time steps available for training the model)

*20. P.12, L303: "PGA≥ 207 m/s²." Please check, these are unphysically large PGA values.*
Thank you very much for identifying this glitch! We revised the whole paper accordingly using [cm/s²] as physical unit and also adapted Figure 1, 5, 10, 11.

*21. P.12, L306-307: "to live in areas which are at peril of a maximum tsunami flow depth of more than two meters". Please check the sentence, it is difficult to read. May I suggest something like "may face tsunami flow depths of two meters or more."?*
Thank you very much for the suggestion. We altered the paper accordingly: Furthermore, more than 600,000 people are anticipated to live in areas which may face tsunami flow depths of two meters or more.

*22. P.12, L308-309: "Waves of up to 20 meters are modeled, and most of the affected people would". I think that "modeled" is too generic, maybe refer to "anticipated scenarios". Similarly, "most of the affected people" is too vague. Please to "exposed people" and specify the %.*
Thank you very much for those suggestions. We revised the paper accordingly and write now: In the considered scenario, waves of up to 20 meters are anticipated, and more than 430 thousand people of the exposed population would be hit by waves higher than 5 meters.

*23. P12, L309-310: "The areas with the largest modeled waves of more than 12 meters only have a small part of the population today, but these will even double in the forecasted 15 years". Similar comment the one before, be more precise and check the sentence which is somehow cumbersome.*
Thank you for sharing this observation. We deleted the sentence accordingly.

*24. P13, Figure 10. Check the units of the PGA.*
Thank you very much for identifying this glitch! We revised the whole paper accordingly using [cm/s²] as physical unit and also adapted Figure 1, 5, 10, 11.

*25. P15, L335: "In this paper, we learned population-related geospatial change trajectories over time and provide". Please check the use of the verb "learned"...we derived? We estimated?*
Thank you for sharing this observation. We revised the sentence as follows: In this paper, we encoded population-related geospatial change trajectories over time in an ML model and provide population forecasts for Peru's capital Lima and Callao…

---

## Author Response (AR3)

**Anticipating a risky future: LSTM models for spatiotemporal extrapolation of population data in areas prone to earthquakes and tsunamis in Lima, Peru**

**March 2024**

Christian Geiß, Jana Maier, Emily So, Elisabeth Schoepfer, Sven Harig, Juan Camilo Gómez Zapata and Yue Zhu

We would like to thank the editor very much for the very valuable and helpful comments. We considered the remaining point and carefully revised the manuscript accordingly. Please find our detailed response to the comment below.

In the response below, the comment of the editor is reported in *italic*. Responses from the authors do not feature any special formatting. Quotations from the manuscript in its revised or in its original form are reported in blue.

**Editor:**

*" The article has undergone thorough revisions from two peers and myself as an editor. It is almost ready to be published in its present form. Nevertheless, there is still a problem with the peak ground acceleration units presented in the upper panels of Figs. 5 , 10 and 11. Because of the resolution of the figure 1, I could not see whether the m/s2 was corrected to cm/s2, please check."*

Thank you very much for this comment. We have carefully revised Fig 5, 10, and 11 and also Fig. 1 to ensure that physically meaningful peak ground acceleration units are documented. Regarding Fig. 1, we provide now a figure with a higher resolution.